# Repellency and Reduction of Offspring Emergence Potential of Some Botanical Extracts against *Sitophilus zeamais* (Coleoptera: Curculionidae) in Stored Maize

**DOI:** 10.3390/insects13090842

**Published:** 2022-09-16

**Authors:** Diaine Cortese, Matheus Moreno Mareco Da Silva, Gisele Silva de Oliveira, Rosilda Mara Mussury, Marcos Gino Fernandes

**Affiliations:** 1Faculty of Agricultural Sciences, Federal University of Grande Dourados, Highway Dourados-Itahum, km 12, Dourados 79804-970, Mato Grosso do Sul, Brazil; 2Faculty of Biological and Environmental Sciences, Federal University of Grande Dourados, Highway Dourados-Itahum, km 12, Dourados 79804-970, Mato Grosso do Sul, Brazil

**Keywords:** bioinsecticide, aqueous extract, maize weevil, stored grains

## Abstract

**Simple Summary:**

*Sitophilus zeamais* is an insect popularly known as maize weevil, usually found in corn, rice, wheat, oats, and barley, among other cereals. However, it is the biggest pest of stored grains which causes economic losses annually with a worldwide distribution. To control this insect, synthetic chemical insecticides are used, which, when misused, may be harmful to human health, animals, and the environment, and may cause resistance in these insect pests, making the insecticides lose efficiency. Therefore, studies that seek new control alternatives are important. In this study we investigated the effects of repellency and reduction of emerged insects of aqueous extracts of five plant species, *S. terebinthifolius*, *L. sericea*, *L*. *tomentosa*, *L*. *longifolia*, and *L*. *nervosa* on *S*. *zeamais*. Aqueous extracts from the leaves of each plant were used to treat corn kernels and thus assess whether they were capable of causing insect repellency. According to our study, the tested species cause repellency of *S*. *zeamais*, in addition to reducing the emergence of insects from the grain. However, this study is preliminary, and more studies are needed to verify that these botanical extracts are ecologically safe.

**Abstract:**

Botanical repellents are, usually, considered safe to control *Sitophilus zeamais*, the main pest of stored maize, as they do not leave toxic residues in food, in addition to having lower costs than chemical insecticides. The aim of this study was to evaluate the repellency potential and the reduction of emerged offspring of botanical extracts, of *Schinus terebinthifolius*, *Ludwigia sericea*, *L*. *tomentosa*, *L*. *nervosa*, *L*. *longifolia*, and use them as botanical insecticides for *S*. *zeamais*. For the repellency test, arenas were set up, containing 50 g of maize kernels exposed to aqueous extract, besides a control treatment. At the center of each arena, 100 insects were released. After 48 h, the proportion of insects in the treated grains was determined. To test the effects of the extracts on insect mating and egg-laying, free-choice and no-choice tests were performed. Insects in both tests remained for ten days for mating and egg-laying. After this period, insects were removed to evaluate the offspring emergence. Sixty days after grain infestation, the number of emerged insects was counted. All selected botanical extracts demonstrated repellent action against *S*. *zeamais*, even after 48 h of application. The *L. nervosa* aqueous extract significantly reduced the emergence of *S*. *zeamais*.

## 1. Introduction

*Sitophilus zeamais* Motschulsky, 1855 (Coleoptera, Curculionidae), popularly known as the maize weevil, is one of the most serious pests of stored products, especially maize (*Zea mays* L.). The maize weevil infests a wide range of cereals and their products, which can lead to weight reduction, depreciation of commercial value, reduction of nutritional value, and secondary causes such as the emergence and dissemination of fungi, due to the increase in moisture in the grain mass caused by the high density of insects [1,2].

Damage caused by *S*. *zeamais* to cereal crops leads to the constant use of synthetic insecticides and this has contributed to the resistance of these insects, making control increasingly difficult. Synthetic insecticides tend to be expensive and have short-lived efficacy [3,4]. In addition, they can present numerous risks to humans, animals, and ecosystems due to high toxicity and residual effects [5,6]. In mind of this, there is a need to investigate the action of botanical extracts to control this insect, with a strategy that provides protection with low cost and high control efficiency.

Identification of plants with insecticidal effects may enable the extraction of organic compounds and, thus, give rise to possible new formulations of synthetic molecules that act as contact insecticides, repellents, and feeding/reproduction suppressors, and be incorporated into integrated pest management (IPM) [7].

The diversity of the Brazilian flora presents an immense potential for the commercial production of organic compounds and needs to be explored, since botanical insecticides are less toxic and cheaper than conventional chemical insecticides. The production of natural chemical compounds is representative for chemical, pharmaceutical, veterinary and plant protection products [8]. In this sense, exploring the action of plant extracts for insect control becomes essential in integrated pest management.

Botanical extracts can be an effective method to control *S*. *zeamais* in stored grains. The use of plants as repellents of storage insect pests is quite old [9]. It is a very viable strategy, as it is, usually, low cost and easily accessible to farmers, minimizing the problems caused by chemical control. In addition to being obtained from renewable resources, they are quickly degraded, act without leaving residues in food, do not damage the ecosystem, and are generally less toxic to humans [10]. In several countries on the African continent, the use of plants for the control of *S. zeamais* has been reported [2,4,9].

Insects have chemical receptors located in various parts of their bodies and these are responsible for assessing environmental conditions, and then fleeing under unfavorable conditions, which leads to repellency behavior [10]. The repellency action is one of the most important properties in the control of insect pests of stored grains with the use of botanical extracts and essential oils. The greater the repellency effect, the lower the infestation, thus, also, favoring a reduction in egg laying and consequently a lower emergence of insects [10,11]. Repellent actions of some plants against *Sitophilus* spp. infesting stored grains have been recently reported [4,12,13,14,15,16].

The action of plant extracts on the insects may occur in different ways, depending on the organic compounds present and their mechanisms of action on the insect, thus, repellent activity is revealed in plant extracts that have compounds which are unpleasant and irritating to insects. However, food deterrence is a disorder associated with sensory mechanisms that ends up causing a reduction in food consumption by insects [17]. Among the compounds responsible for these mechanisms of action, are phenolic compounds, flavonoids, tannins, and alkaloids [18]. Although the use of synthetic insecticides is still the main way to control *S. zeamais*, many authors have revealed the resistance of populations of this insect to several chemical insecticides [19,20,21], which gives encouragement to seek new control methods. In addition, according to the scientific literature, the method of using botanical extracts is less likely to develop resistance in insects [11,12,13,14,15].

This study aimed to determine the effects of botanical extracts of five plant species *Schinus terebinthifolius*, *Ludwigia sericea*, *L. tomentosa*, *L. nervosa*, and *L. longifolia* in reducing offspring emergence and repellency potential of adult insects.

## 2. Material and Methods

### 2.1. Selection and Preparation of Botanical Extracts

Expanded leaves of *Schinus terebinthifolius*, Raddi., *L. tomentosa* (Cambess.) H. Hara, *L. longifolia* (DC.) H. Hara, *L. sericea* (Cambess.) H. Hara, and *L. nervosa* (Poir.) H. Hara were collected from highway surroundings in the city of Dourados, MS. The collection area was between the Atlantic Forest and the Cerrado which was beginning to regenerate secondary succession, (22°11′54.92″ S, 54°46′52.15″ O). The plant species were deposited at the Herbarium of the Federal University of Grande Dourados-UFGD, with the following registration numbers: 6391-*L. tomentosa*, 6389-*L. longifolia*, 6388-*L. sericea* and 6390-*L. nervosa*, 6473-*S. terebinthifolius*. The collection of the botanical material was authorized by the Brazilian National Research Council (CNPq)/Council of Genetic Heritage Management (CGEN/MMA), under the number A9ECAC6 [18].

Aqueous extracts were obtained from leaves of five plant species (*L. sericea*, *L. tomentosa*, *L. nervosa*, and *L. longifolia*). We chose these species based on references regarding their promising use as insecticides [18].

Leaves of each plant species were sanitized in the laboratory and then dried in a forced air oven for 72 h at a maximum temperature of (45 °C ± 1 °C). After this period, the leaves were ground in a micro knife mill (MA048). The resulting powder was placed in a capped glass vial, at room temperature.

To prepare the aqueous extracts of each species, 5 g powder from the leaves of each species was weighed on an analytical balance and placed separately in a glass beaker containing 50 mL distilled water, thus obtaining a final concentration of 10%, which is the usual used concentration of the extracts of these plants to control insect pests [18]. After 24 h, extracts were filtered through filter paper for use in the tests.

### 2.2. Mass Breeding of S. zeamais

Insects used in the experiments were from mass breeding in the laboratory, kept in 1 L plastic pots and fed whole maize grains free of insecticides and pests, under controlled temperature (27 ± 2 °C), relative humidity (70 ± 5%), and photoperiod (LD 12:12). The same conditions of temperature, humidity, and photoperiod were used in repellency tests of plant extracts and in the tests of the extracts effects on the *S. zeamais* number of emerged progeny, both in the choice test and the no-choice test.

### 2.3. Repellency Test

In the repellency test, five treatments were evaluated, which consisted of four aqueous botanical extracts from the plants *S. terebinthifolius*, *L. sericea*, *L. tomentosa*, *L. nervosa*, at 10% concentration and treatment with distilled water as a control.

In the experiments carried out in this research, no positive control (reference insecticide) was used, because synthetic chemical insecticides are usually used as the positive control [22], and these insecticides are used through fumigation, purge or thermo-nebulization techniques. Thus, in the case of using this positive control, there would have been distortions in the repellency effects of the botanical extracts.

To set up the test, 50 g maize from a commercial (DKB 290 pro3) cultivar free of insecticides and pests was dipped in each treatment and lightly shaken by hand until all the grains had been properly covered by the filtrates. Then, the grains were sieved with the aid of a mesh size sieve (125 mm–20 µm) and deposited on filter paper to absorb the aqueous extracts and drying at room temperature. After that, 50 g maize was placed in Petri dishes, labeled, and kept in white square plastic trays (25 × 35 cm), used as arenas for the free choice of insects, for feeding and egg laying, in the different treatments. One-hundred insects deprived of food for 24 h, non-sexed, and aged between 7 and 14 days were released at the center of each tray [23,24]. To seal the trays, in order to prevent insect escape, PVC plastic film was used, in two layers, duly sealed on the edge of each tray with adhesive tape (Figure 1). Micro-holes were made in the plastic film with number 2 (0.46 mm in diameter) entomological pin to allow good airflow.

As an evaluation parameter to determine the repellency of the treatments, the number of insects in each Petri dish was counted, after 48 h. Thus, the proportion of insects present in the tested extracts was determined [23]. For this test, ten replicates were used for each treatment.

### 2.4. Test of the Effects of the Plant Extracts on the S. zeamais Number of Emerged Progeny

For the evaluation of plant extracts on the insects mating and egg-lying, two tests were performed: one test with multiple choice and another test without choice. For these tests, 10% aqueous extracts of the plants *S. terebinthifolius*, *L. sericea*, *L. tomentosa*, *L. nervosa*, *L. longifolia* were evaluated as treatments, and a control (distilled water). The extract effects on the F1 progeny of *S. zeamais* were evaluated. For both tests, thirty grams of maize was dipped in the filtrates of each treatment and then this was sieved and placed to dry on filter paper at room temperature and, once dry, deposited in Petri dishes. To evaluate the no-choice test, 20 adult *S. zeamais* individuals, aged 7 to 14 days, were used to infest 30 g of corn in petri dishes. These adults had been previously deprived of food for 24 h.

For the multiple-choice test, a sample of each treatment was randomly placed inside white plastic trays (25 × 35 cm), used as arenas, and 120 *S. zeamais* individuals were released at the center of the tray, for free choice of grains. After that, the trays were sealed with plastic PVC film (28 × 30 cm), with two layers and with adhesive tape, to prevent insects from escaping [23]. Insects in both tests remained ten days for mating and egg-laying on the grains (Figure 2). After this period, the insects were sieved from the grains and discarded in order to evaluate the offspring emergence. Sixty days after grain infestation, the number of insects having emerged from each sample in both tests was counted [25]. For the two tests, ten replications were used for each treatment.

### 2.5. Statistical Analysis

Experiments were carried out in a completely randomized design, with five treatments for the repellency test and six treatments for the test of plant extracts on the insect mating and egg-lying, using ten replications for each test.

Data were initially tested for normality by the Shapiro–Wilk test. To confirm the normal distribution, data referring to repellency and emergence were transformed into the square root of (x + 0.5), in all experimental tests. Then, variables were tested by analysis of variance (ANOVA) and Tukey’s test with 1% probability, to compare the proportion of individuals that visited the treated area and the percentage of repellency between treatments. For the other tests, the same statistical analyses were applied. The data were analyzed using the R platform.

## 3. Results

The experiments showed that maize grains treated with plant extracts from *S. terebinthifolius*, *L. sericea*, *L. tomentosa*, and *L. nervosa* affected the behavior of *S. zeamais*, as they were significantly less visited than the grains of the control treatment (Table 1).

The high number of insects that avoided contact with grains treated with plant extracts was confirmed with the mean test, demonstrating a significant repellency for *S. zeamais*. The control treatment had a mean value of 52.70 insects found in the grains, about 50% of the insects used for infestation (Table 1).

Although the plant extracts did not differ significantly, the *L. nervosa* plant extract showed the lowest mean value, 3.5 insects found on the treated grains, after 48 h of exposure (Table 1).

For the multiple-choice test, which evaluated the insecticidal action of botanical extracts against the F1 progeny of *S. zeamais*, the control treatment presented the highest emergence of insects, with a mean value of 28.60 insects, differing from the other treatments, except for *L. tomentosa* (Table 2).

Botanical extracts of *L. sericea*, *L. longifolia*, *S. terebinthifolius*, and *L. nervosa* plants differed statistically from the control group (Table 2). Among these there were no statistical differences in the emergence of insects, however the botanical extract of *L. nervosa* resulted in the lowest number of insects emerged, presenting a mean value of 11.50 insects (Table 2).

The control group showed the highest mean number of emerged insects, with a value of 38.00 insects, in the no-choice test, statistically differing from *L. sericea* and *L. nervosa* (Table 3).

The botanical extract of *L. longifolia* had a mean value of 32.30 emerged insects, not differing statistically from the control group, nor from the extracts of *S. terenbithifolius* and *L. tomentosa* (Table 3). The botanical extract of *L. nervosa* had the lowest mean value of emerged insects (8.9 insects) among the other botanical extracts, however, this botanical extract did not differ statistically from *L. sericea* and *L. tomentosa* extracts, with their respective mean values of 15.20 and 20.70 insects (Table 3).

## 4. Discussion

The tested aqueous extracts from plants *S. terebinthifolius*, *L. sericea*, *L. tomentosa*, *L. nervosa*, and *L. longifolia*, proved to be effective against *S. zeamais* in stored maize grains, with regard to their repellency. We can consider the possibility that this effect of repellency was simply due to the fact that the natural odor of the maize was masked by the extracts, resulting in the weevils being unable to locate the grains. However, this fact would still confirm the repellency effect of these substances. The selection of these plants to obtain the extracts aiming to investigate their repellency and insecticidal effects, was based on phytochemical and insecticidal activity studies developed by ref. [18], which demonstrated great insecticidal potential of these plant extracts against *Plutella xylostella* (L., 1758).

The results of the repellency test pointed to a significant difference between the extracts and the control treatment after 48 h. Statistical analyses showed that botanical extracts of *S. terebinthifolius*, *L. sericea*, *L. tomentosa*, and *L. nervosa*, induced a repellency behavior, as the insects avoided the maize grains treated with these extracts. This shows that botanical extracts of these plants can reduce *S. zeamais* infestation in stored maize grain, by insect repellency.

All selected botanical extracts had repellent action against *S. zeamais* at the tested concentration. A low number of insects in areas with treated grains was found, even after 48 h after application of treatments. It can be assumed that the botanical extracts contain volatile compounds with a long duration after application. Plant extracts must act as repellents driving the insects away due to their odor and/or taste. As arthropods tend to flee from areas with a pungent odor [26], or by the unpleasant taste, the repulsion of *S. zeamais* by these extracts was possibly accomplished through the stimulation of olfactory or gustatory receptors [4].

Active compounds such as alkaloids, flavonoids, saponins, phenolic compounds, and tannins are present in many plants of *Schinus* spp. and these can disrupt olfactory receptors so that insects cannot detect their host [2,3,4,5,6,7,8,9,10,11,12,13,14,15]. This may be the reason why there was a greater number of insects in the control group after 48 h.

The control treatment had 50% insects in its grains, and the rest of the insects, despite not being in the grains of the control group, were in areas of untreated grains, which confirms the repellency by the extracts tested in this study. In fact, it is common for these insects to remain in areas not treated with plant extracts, since they avoid areas treated with the extracts with irritating effects [4], confirming that, in our study, this is a repellency behavior.

Some studies that could be carried out in complement to this one are the ones required to assess the lethal and sublethal effects of the botanical extracts tested here to elucidate their insecticidal activities. It is very well known that the use of *Schinus molle* L. essential oil is effective in killing *S. zeamais* [11]. The oil of this plant showed an insecticide effect of contact and repellency, and this effect can lead to different consequences for the insect, such as repellency, inhibition of egg-laying, growth, feeding, morphogenetic changes, hormonal system, sexual behavior, and mortality in the adult or immature stages [11]. This demonstrates that *S. molle* has insecticidal potential, which, in fact, we confirmed in our studies in that the use of the extract of *S. terebinthifolius*, a plant of the same genus as the study cited above, provides evidence of repellency in maize grains against *S. zeamais*.

In the multiple-choice test, *S. terebinthifolius* extract presented the second lowest mean value of emerged insects, after sixty days. However, when the insects did not have the chance to choose, the botanical extract of this plant did not differ significantly from the control group, in relation to the number of insects emerged. We concluded that the aqueous extract of *S. terebinthifolius* does not affect the development of insects of the *S. zeamais* species in maize grains.

When *S. terebinthifolius* aqueous extract is applied to stored wheat grains, it seems to have an effect on larvae and egg-laying of *S. zeamais*, as it causes low hatching of insects, with 61% efficacy [10]. Since we proved that the repellency effect causes the reduction in the number of insects emerged after sixty days of using the aqueous extract in stored maize grains, we can affirm that this extract results in a low reinfestation of insects in the grain mass, favoring a smaller loss of these grains.

Research has shown that the mortality of *S. zeamais* adults who were fed the extract of leaves of *S. terebinthifolius*, in addition to a strong deterrent effect of the intestine and proteolytic enzymes [27], may be due to the presence of flavonoids in this plant. The rejection of diet by insects, caused by the extract of this plant, suggests that its use may have a protective action of grains.

The effects of repellency and reduction in the number of emerged adults of *S. zeamais* observed for *S. terebinthifolius* may have been due to its very rich chemical composition. This plant has a variety of bioactive substances in its composition, such as: terebinthone, hydroxymasticadienoic acid, terebinthifolic acid, and ursolic acid. In addition to these compounds, this plant species has other constituents, such as tannins, flavonoids, terpenoids, among others [28]. In a study carried out to identify the chemical composition, biological properties, and toxicity of *S. terebinthifolius*, the high opulence of compounds present in this plant was demonstrated [29].

The effectiveness of using any plant for protection against insects can be measured by their mortality rate, the target organism, and sublethal effects, such as the effects on their development, reproduction, and behavior [23]. The behavior change is evidenced after the use of *L. nervosa* extract, by the lower number of insects in the treated area after 48 h, which proves the repellency effect caused by this aqueous extract on *S. zeamais*.

Regarding the potential killing effect of the extracts, we noticed that after ten days the insects were in contact with the treated grains and using them for food, no insect mortality was seen after this period. However, the efficacy of sublethal effects of the aqueous extract of *L. nervosa* was proved, due to the very low number of insects emerged in the no-choice test. Although *L. nervosa* extract did not cause adult mortality of *S. zeamais*, there was an evident effect on breeding and egg-laying behavior, due to the smaller number of insects emerged in the no-choice test.

Studies carried out by ref. [18] suggest that phytochemicals compounds change the behavior, breeding, and egg-laying of *S. zeamais* because of the large amount of phenolic compounds (312.4 mg/g), flavonoids (188.8 mg/g), tannins (34.4 mg/g), and alkaloids (12.5 mg/g) in the extract of *L. nervosa*. In this cited study, the extract of *L. longifolia* showed a greater emergence of insects for the no-choice test, in relation to the other botanical extracts (*L. nervosa*). This may be directly related to the lower amount of phenolic compounds (289.7 mg/g), flavonoids (144.9 mg/g), tannins (30.9 mg/g), and alkaloids (10.4 mg/g), presented by *L. longifolia*.

In the scientific literature, there are few studies that have proved insect repellency and insecticidal action of the plant species tested herein. On the same species of *Ludwigia* ssp., it was found that *L. tomentosa*, *L. longifolia*, *L. sericea* showed excellent results for the control of *P. xylostella*; although *L. nervosa* has higher levels of phenolic compounds, flavonoids, tannins, and alkaloids, it does not perform well for the control of this insect [18].

After analyzing our results and comparing them with the results already published on the effect of plant extracts, we concluded that different insect pest species may have different behavior to the same plant extract, but this does not affect its effectiveness as an alternative tactic for insect pest control. For example, extracts of *Commiphora myrrha* (L.), *Acorus gramineus* (Soland.), and *Kaempferia galanga* (L.) are known to have opposite effects on *S. zeamais* and *Plodia interpunctella* (Hubner, 1813), since *P. interpunctella* prefers extracts that are repellent to *S. zeamais* [13]. Therefore, studies with different plant extracts should be conducted and evaluated on weevils, in order to find better insecticidal extract for controlling these insect pests.

Importantly, due to the strong repellent effect presented by botanical extracts of *S. terebinthifolius*, *L. sericea*, *L. tomentosa*, *L. longifolia*, and especially *L. nervosa*, these products can be used as a strategy in integrated pest management (IPM) in stored grains. These botanical extracts can alleviate the problems associated with the use of synthetic chemical insecticides, as they can reduce storage costs, since they are low-cost products that can even be produced on the farm [2,30]. As these products are extracted from renewable sources, a greater use of these products will be beneficial for environmental conservation on the planet. Besides, they present little or no risk to natural enemies, or to humans and animals, and have no residual effects, and also reduce the possibility of development of resistance on stored grain insect pests. Furthermore, the use of botanical extracts can be very promising in grain warehouses, since powders and extracts are easy to apply while the grains remain fresh, clean, and attractive for marketing after application [30].

## 5. Conclusions

Botanical extracts of *S. terebinthifolius*, *L. sericea*, *L. tomentosa*, *L. nervosa*, and *L. longifolia* have repellent action against *S. zeamais* when applied to the grain mass, since these insects avoid feeding on the treated grains. Additionally, the *L. nervosa* aqueous extract has an excellent effect on reducing the emergence of *S. zeamais* adults on maize grains.

## Figures and Tables

**Figure 1 insects-13-00842-f001:**
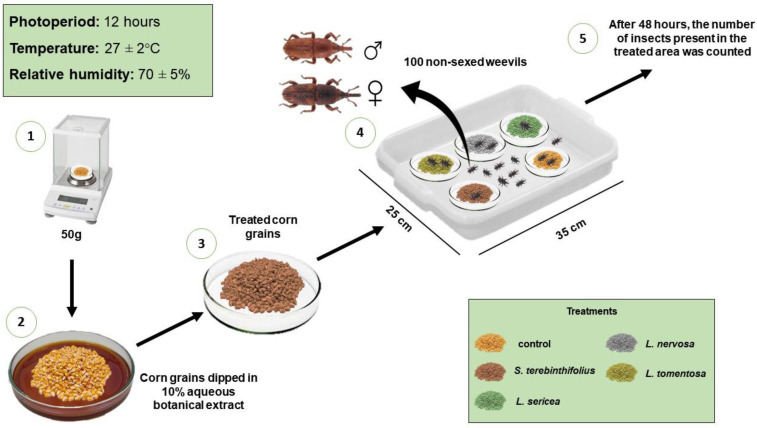
The schematic representation of methodology of the test to evaluate the repellency of botanical extracts to *Sitophilus zeamais*.

**Figure 2 insects-13-00842-f002:**
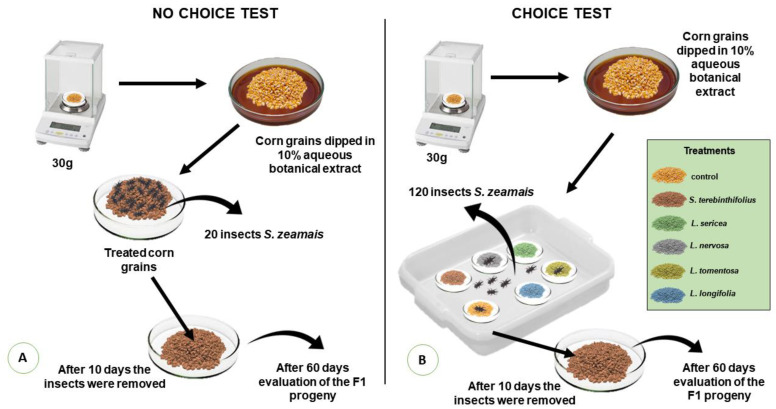
A schematic representation of the methodology used for the test of plant extracts on insect mating and egg-lying. Subfigure (**A**) represents the methodology used in the no choice test for oviposition of *S. zeamais* in corn kernels treated with botanical plant extracts. Subfigure (**B**) represents the methodology used in the free-choice test for oviposition of *S. zeamais* in corn kernels treated with botanical plant extracts *S. terebinthifolius*, *L. sericea*, *L. tomentosa*, *L. nervosa*, *L. longifolia* in both tests.

**Table 1 insects-13-00842-t001:** Repellency test of four plant extracts (*Schinus terebinthifolius*, *Ludwigia sericea*, *Ludwigia tomentosa*, *Ludwigia nervosa*) and a control (water) on *Sitophilus zeamais*. Dourados, 2021.

Treatments	Number of Insects
Control	52.70 (44.71–60.68) a
*Schinus terebinthifolius*	6.90 (3.76–10.03) b
*Ludwigia sericea*	6.0 (3.19–8.80) b
*Ludwigia tomentosa*	4.80 (1.98–7.61) b
*Ludwigia nervosa*	3.50 (1.25–5.74) b
F	73.06
P	<0.0001
DF	4
CV (%)	25.61

Mean values followed by different letters are significantly different from each other at the significance level α = 0.001, compared by Tukey’s test. CV: Coefficient of variation. N = 10. P: Probability. DF: Degree of Fredow. F: Value of Significance.

**Table 2 insects-13-00842-t002:** Number of insects emerged after 60 days of *Sitophilus zeamais* infestation (choice test), in thirty grams of maize treated with 10% aqueous extract of five plants and one control (water), Dourados, 2021.

Treatments	Number of Insects Emerged
Control	28.60 (18.90–38.29) a
*Ludwigia tomentosa*	15.80 (11.09–20.50) ab
*Ludwigia sericea*	13.60 (9.38–17.81) b
*Ludwigia longifolia*	12.40 (7.71–17.08) b
*Schinus terebinthifolius*	12.50 (7.43–17.56) b
*Ludwigia nervosa*	11.50 (7.65–15.34) b
F	5.64
P	<0.0003
DF	5
CV (%)	24.51

Mean values followed by different letters are significantly different from each other at the significance level α = 0.001, compared by Tukey’s test. CV: Coefficient of variation. N = 10. P: Probability. DF: Degree of Fredow. F: Value of Significance.

**Table 3 insects-13-00842-t003:** Number of insects emerged after 60 days of *Sitophilus zeamais* infestation (no-choice test), in thirty grams of maize treated with 10% aqueous extract of five plants and one control (water). Dourados, 2021.

Treatments	Number of Insects Emerged
Control	38.00 (25.88–50.11) a
*Ludwigia longifolia*	32.30 (22.19–42.40) a
*Schinus terenbithifolius*	24.20 (18.25–29.44) ab
*Ludwigia tomentosa*	20.70 (12.79–28.60) abc
*Ludwigia sericea*	15.20 (9.00–21.39) bc
*Ludwigia nervosa*	8.9 (7.00–10.79) c
F	11.48
P	<0.0001
DF	5
CV (%)	22.57

Mean values followed by different letters are significantly different from each other at the significance level α = 0.001, compared by Tukey’s test. CV: Coefficient of variation. N = 10. P: Probability. DF: Degree of Fredow. F: Value of Significance.

## Data Availability

The data presented in this study are available in the article.

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
