# Peer review of "Repellency and Reduction of Offspring Emergence Potential of Some Botanical Extracts against *Sitophilus zeamais* (Coleoptera: Curculionidae) in Stored Maize"

_insects, 2022, doi:10.3390/insects13090842_

Round 1

Reviewer 1 Report

I feel additional studies need to be performed to accurately make the conclusions stated in the manuscript. The assays are very simplistic and cannot support your claims. I have made comments in the manuscript. 

Author Response

On behalf of all authors, all questions were useful for the improvement of the work and were directly considered in the writing of the article. "Please see the attachment."

Reviewer 2 Report

Dear authors,

This paper deals with the evaluation of repellent potential to adults and effects on progeny of Sitophilus zeamais (corn weevil) after treatment of maize grains with five plant extracts.

I have some comments that may benefit the manuscript.

Lines 87-90: These lines should be transferred to the previous text. Here you would better focus on the aim of the current paper which is not strictly related to resistance.

Line 91: Please make more specific the studied potential of the extracts. It seems that the study dealt with the repellent potential on adults and effect on progeny. Please clarify and amend these lines accordingly.

Line 128: It seems that with “negative control” you mean reference product used as positive control. Please clarify and amend the text accordingly.

General comment: The adult repellence was evaluated only in multiple choice test in the same closed arena, where the repellent action (through the vapor) of each individual extract may affected by the action from the other ones. It seems that a non-choice test would be necessary to support the effect of each extract, like in the case of progeny effects. Could you please explain why the multiple-choice test set up was not a bias in terms of the evaluation of the repellent effect for each extract?

Line 151: It is speculated that with the two experiments presented in the paragraph, you studied mating and egg-laying. However, the only measurement was the number of emerged progeny, whereas no mating status or ovipositioning were recorded. You may reconsider the title of the paragraph and elsewhere in the manuscript to address that the tested effect was on offspring emergence, not on mating and egg laying. This should be also considered in the title of the manuscript: e.g. by changing to “Repellency and offspring emergence potential of botanical extracts…”.

Lines 285-286: What was the killing effect and in which trial?

Line 299: In which trial no mortality was observed?

Generally, for each extract in each trial any killing effect that was observed should be reported in the manuscript (also the non-killing effect) and should be considered in the discussion.

Author Response

(The authors gave the same response as above.)

Round 2

Reviewer 1 Report

Please see the recommended edits in the attached document. 

Author Response

Reply to reviewer
Regarding the title of the article, we did not change it because the other reviewer asked 
that it be that one. Indicating that the studied effects of the botanical extracts was in the 
emergence of S. zeamais.
1-How does this confirm the repellency of the extracts?
Repellent action occurs through contact or inhalation, as the insect identifies 
irritating substances and avoids the treated area.
2- An interesting addition to your study would be to place all the insects on the treated 
grain and see if they leave. That would validate your repellency claim. 
If we take into account that the insects were 24 hours without food and another 48 
hours after the application of the products, then a total of 72 hours. Even so the 
insects would not seek food and their host for oviposition etc, I believe it can be 
assumed that there is a repellency. We did a preliminary repellency test with the 
same extracts for Tribolium castaneum and this insect was found even in a higher 
proportion in areas treated with the botanical extracts than in corn kernels without 
treatment with the botanical extracts. So for S. zeamais these extracts have a 
repellent effect.
3- What are the "irritating effects"?
Plants with insecticidal effects can act as insect feeding inhibitors or impediments
growth, development, reproduction and behavior. Action on the central nervous 
system. Anti-food agent. Activity on target organ or molecule. In this case, they work
hindering growth and development, interfering with cellular metabolism.
Depending on the concentration used, some extracts can reduce the viability of eggs, 
nymphs, larvae and pupae. The reduction in the number of eggs and the inhibition 
of oviposition are important effects of plant extracts on insect reproduction. Other 
substances act by contact, or that is, they act on and are absorbed by chitin and
exoskeleton or through the airways (action fumigant), can be useful for pest control
that attack food in warehouses and silos.
Reference. 
Corrêa, J.C.R., & Salgado, H.D.N. (2011). Insecticidal activity of plants and 
applications: review. Brazilian Journal of Medicinal Plants, 13, 500-506.
4- If the extract has the ability to kill the insect within the grain, what are the effects on 
humans or livestock consuming the treated grain?
We do not know the effects on humans, but preliminary studies are needed to 
identify the effective insecticide. If there are good results, then studies can be 
improved and deepened, for possible effects on mammals.
5- What does this mean?
Due to the ingestion of substances (Lectin SteLL) it is useful as an additive or 
synergistic agent capable of reducing the aptitude of pests, affecting the conversion 
of food into biomass. Proteins in this class can also affect insect metabolism.
Modulating the activity of intestinal enzymes. deterrent effect post-ingestion: That 
these lectins present are capable of altering the activity of enzymes such as amylase, 
protease, or Trypsin.
6- What does this mean?
7- What are the effects of these compounds? Are they good or bad for the insects?
8- Again, what are the effects of these compounds on the insect?
Answer the questions below.
Flavonoids are responsible for the reduced growth of larvae [51] and pupal survival 
[52], impaired feeding, digestion inhibition and the release of free radicals [53]. 
Flavonoids such as quercetin 3-arabinoside, quercetin 3-glucoside and quercetin 3-
rutinoside have already been identified in some species of Ludwigia [54] and found 
to be able to act as phagodeterrents, depending on the concentrations used [35].
Tannins are another class of compounds with anti-food effects [55]. The alkaloids 
observed may also interfere with neuroendocrine control by inactivating 
acetylcholinesterase in larvae, causing neurotoxicity [56], in addition to a decrease 
in weight and increased mortality. This is indicated by a significant decrease in 
proteins, glycogen, lipids and the activity of the digestive enzyme α-amylase [36].
Reference in this article
Ferreira, E.A., de Souza, S.A., Domingues, A., Da Silva, M.M.M., Padial, I.M.P.M., 
de Carvalho, E.M., ... & Mussury, R.M. (2020). Phytochemical screening and 
bioactivity of Ludwigia spp. in the control of Plutella xylostella (Lepidoptera: 
Plutellidae). Insects, 11(9), 596.
The vast majority of works in the literature that refer to terpenoids superiors, refer 
to observations of activities such as growth inhibitors or retarders, maturation 
damage, reduced reproductive capacity, appetite suppressants, which may drive 
predatory insects to death by starvation or direct toxicity.
Reference in this article
Viegas Junior, C. (2003). Terpenes with insecticidal activity: an alternative for 
chemical control of insects. Química Nova, 26, 390-400.
9- How are they low-cost? How and why would a crop producer manufacture the 
extracts?
There are many scientific articles that report the use of botanical chemical 
insecticides to control pests of stored grains mainly in African countries, where 
farmers do not have the financial resources to obtain expensive synthetic chemical 
insecticides that are often manufactured in rich and well-off countries. developed. 
which I mentioned in this work. The cost is cheaper because the botanical insecticide 
manufacturing process does not need expensive raw materials or an 
industrialization process that would increase the cost.
For other corrections, questions and statements were modified in the body of the scientific 
article's text.
We thank you immensely for each correction, as it was of paramount importance for the 
quality and improvement of the article. 

Reviewer 2 Report

Dear authors,

Thank you for considering my comments. However, the following two issues remain still open.

In line 158 and in other parts of the manuscript it is speculated that mating and egg-laying of Sitophilus zeamais was evaluated as a result of the plant extracts performance. However, the only measurement was the number of emerged progeny, whereas no mating status or ovipositioning were recorded. Hence, the title of the paragraph 2.4 should be reconsidered stating only the effect on progeny. In the text of this paragraph you may just add some of the speculations for potential effects on mating and egg laying that are mentioned in your answer. I propose to remove from the whole manuscript including the abstract and figures any direct statement for effects on mating and egg laying.

In your answer regarding the potential killing effect of the extracts you say that ”…after ten days the insects were also in contact with the treated grains and using the for food, no insect mortality was seen after this period”. This information is quite useful and should be added in the manuscript clarifying also the trials the trials that is referring to.

Author Response

Reply to viewer
Dear authors,
Thank you for considering my comments. However, the following two issues 
remain still open.
In line 158 and in other parts of the manuscript it is speculated that mating and 
egg-laying of Sitophilus zeamais was evaluated as a result of the plant extracts 
performance. However, the only measurement was the number of emerged 
progeny, whereas no mating status or ovipositioning were recorded. Hence, the 
title of the paragraph 2.4 should be reconsidered stating only the effect on 
progeny. In the text of this paragraph you may just add some of the speculations 
for potential effects on mating and egg laying that are mentioned in your answer. 
I propose to remove from the whole manuscript including the abstract and figures 
any direct statement for effects on mating and egg laying.
Answer: it was budget in the text. Thanks for contributing to the improvement of 
the article.
In your answer regarding the potential killing effect of the extracts you say that 
”…after ten days the insects were also in contact with the treated grains and using 
the for food, no insect mortality was seen after this period”. This information is 
quite useful and should be added in the manuscript clarifying also the trials the 
trials that is referring to.
Answer: it was budget in the text. Thanks for contributing to the improvement of 
the article.